# Knowledge of Sexuality and Contraception in Students at a Spanish University: A Descriptive Study

**DOI:** 10.3390/healthcare10091695

**Published:** 2022-09-05

**Authors:** Sebastián Sanz-Martos, Isabel M. López-Medina, Cristina Álvarez-García, Lucía Ortega-Donaire, María E. Fernández-Martínez, Carmen Álvarez-Nieto

**Affiliations:** 1Department of Nursing, Faculty of Health Sciences, University of Jaen, 23071 Jaen, Spain; 2061 Health Emergency Centre, Andalusian Health Service, 23071 Jaen, Spain

**Keywords:** unwanted pregnancy, primary prevention, knowledge, young adult

## Abstract

Youth is a period characterized by impulsiveness and risk-taking. This population often feels invulnerable and has a strong motivation to seek out their identity. These characteristics make it a risky age period for an unwanted pregnancy. This study aimed to investigate the level of knowledge about sexuality and contraception among nursing students at the University of Jaen. The students completed a 16-item questionnaire assessing their knowledge level about sexuality and contraception. A bivariate analysis was performed using the Kruskal–Wallis H and Mann–Whitney U tests. Later, an ordinal logistic regression model was generated. The sample consisted of 130 participants, with an average age of 20.36 years. The level of knowledge about sexuality and contraception was moderately high, 10.38 points out of 16. The factors associated with the probability of accordingly presenting a high level of knowledge were being a woman, having studied in a public institution, not having used any contraceptive method in the first sexual experience, and wishing to use a contraceptive pill in future sexual relations. In conclusion, the knowledge level about sexuality and contraception is high, although it is necessary to assess whether this level of high knowledge translates to the use of contraceptive methods in future sexual experiences.

## 1. Introduction

The World Health Organization (WHO) defines “youth” as those aged between 15 and 24 years [1]. Youth is a period characterized by impulsiveness and risk-taking. This population often feels invulnerable and has a strong motivation to seek out their identity. During youth, some personality traits may appear that lead them to feel observed and judged in every action they carry out, along with feelings of invulnerability and feeling different from others and ignoring the risk of their actions when they contradict their desires or the actions they seek to carry out. Sexual experiences present some remarkable characteristics, as within the friendship group, it is considered an evolutionary achievement to achieve success on a sexual level, and it provides the attainment of immediate pleasure, with a low probability of any adverse consequences, which reinforces the personality characteristics present in young people of invulnerability or low perception of risk [2,3,4]. The combination of all the above is associated with a need for experimentation (which is a form of learning) during the age period of adolescence and youth and, in turn, can be related to early sexual initiation with the consequent risk of unwanted pregnancies, abortions, and sexually transmitted infections [2]. In recent years, the age of sexual debut has remained stable globally, although with a slight increase for the young age group [5,6,7]. There were notable differences between developed societies such as the USA, with a median age of 15.4 years [8], and Nigeria, with a median age of 13.12 years [9]. However, research undertaken in Spain found that there has been a decrease in the age of sexual incitation with penetration; according to a previous multicenter study with nursing degree students [10], this age was 16.55 for youth in the 18–24 age group, lower, in the 16–18 age group (15.66 years), 16.73 for the 19–21 age group, and 17.53 for the 22–25 age group [11]. Additionally, a low contraceptive prevalence rate was observed in the Spanish Contraception Society’s survey in 2020 [12], where 37.1% of women aged 15–19 and 24.5% aged 20–24 did not use any contraceptive method during their most recent sexual experience. Spain has a fertility rate of 4.68 pregnancies per 1000 women in the 15–19 age group in 2021 and 20.41 pregnancies per 1000 women in the 20–24 age group [13].

A young person’s experience of sexuality may be marked by myths or consequences reinforced by the misinformation they possess and by the shame they feel when seeking information about their feelings in the face of changes, making it difficult to experience healthy sexuality [14,15]. The WHO defines sexuality as a fundamental dimension consisting of a set of biological, psychological, and socio-cultural characteristics that constitute the person [16]. Young people are physically ready to experience their sexuality; however, they may not be psychologically and emotionally prepared for the new challenge [2,3,4]. The experience of healthy and risk-free sexuality is one of the main objectives of the WHO, seeking to achieve high levels of sexual and reproductive health, for which access to quality sources of information is established as a key element for building a solid and sufficient body of knowledge to live the new stage without associated risks [17]. The main sources of information on sexuality and contraception among young people are the Internet, parents, friends, and the media [11,12,18,19]. Previous research on university students has found a moderate level of knowledge about sexuality and contraception, which increases significantly after undergraduate training [11,20,21].

This research aimed to determine the main sources of information, level of knowledge about heterosexuality and contraception, and contraception use rate among university students in Spain. Knowing the characteristics of the sexual and contraceptive behavior of university students, their main sources of information, and their level of knowledge about heterosexuality and contraception will aid the implementation of effective educational programs for the prevention of unwanted pregnancy, addressing them through the sources used and focusing on the main gaps in knowledge.

## 2. Materials and Methods

### 2.1. Design

The study was designed as a cross-sectional study.

### 2.2. Sample and Setting

The participants in the study were second-year students of the Degree in Nursing (University of Jaen) who were studying the subject of *Nursing in childhood and adolescence*, with ages between 18 and 24 years. A convenience sample was used due to the accessibility of the sample of students in the second year of the Nursing Degree. Based on previous research [3,11], a sample size calculation was established to detect a difference of 1.5 points in knowledge level, a standard deviation of 2.5 points, a confidence level of 95%, and a power of 80%. Therefore, the minimum sample size was established at 90 participants.

### 2.3. Data Collection

Between February and March 2016, during the theoretical and practical sessions, students responded to an ad hoc questionnaire designed for this research. The instrument consisted of three distinct sections: the first part consisted of demographic questions, the second part asked questions about their pattern of sexual behavior, and the third part assessed their level of knowledge about heterosexuality and contraception. For most of the questions about demographics and sexual behavior, the participants were asked to choose one of the proposed options. However, for other questions, a variety of answers other than those listed were possible; thus, a free answer choice was added where the participants could write their answers. Subsequently, all the responses were analyzed and re-coded into qualitative variables. The questionnaire is available in Appendix A:
Socio-demographic variables: sex, age, type of school where they studied, source of information used to obtain information on sexuality and contraception, source of information demanded, and self-perception of the level of knowledge they have about heterosexuality and contraception.Sexual behavior: Previous sexual experience (yes/no), age of initiation, use of contraception during first sexual experience, contraceptive method used during first sexual experience, and contraceptive method of choice for future sexual relations.Knowledge about heterosexuality and contraceptive methods: To measure the construct knowledge level, an initial bank of 23 dichotomous (true/false) response items pertaining to knowledge about heterosexuality, knowledge about male condoms, and knowledge about hormonal contraceptive methods was generated. The clarity with which the items were written and the relevance of the items for assessing the study construct was evaluated by a committee of five national experts in the studied construct. The validity of the content was assessed based on the degree of agreement among the experts on the relevance and clarity of the items selected to measure the construct, using Aiken’s V and selecting those items that obtained a value greater than 0.7. Following the experts’ assessment, eight items were redrafted because they were not considered clear, four were deleted at the request of three experts who did not consider them relevant for measuring the construct, and three were deleted because they did not reach the limit value for the degree of agreement on the relevance for measuring the construct. The final instrument contained 16 items divided into three dimensions: knowledge about heterosexuality (six items), knowledge about male condoms (five items), and knowledge about hormonal contraceptive methods (five items).For each question answered correctly by the participants, one point was added to the total score, and this score was then corrected using the formula for the correction of randomness based on the number of answer options (Correct responses-errors/number of answer options-1), so that in our case it is simplified to correct responses-errors, obtaining the final corrected score on a discrete quantitative scale. From this final score, a re-coding of the participants’ scores was carried out, obtaining a qualitative ordinal variable with three levels of knowledge:
○High Knowledge: Correct answers ≥ 75%;○Low knowledge: Correct answers (≥50%, <75%);○Null knowledge: Correct answers < 50%.;


### 2.4. Statistical Analysis

A descriptive analysis of the qualitative variables was carried out, obtaining their frequency and percentage distribution. Measures of central tendency and dispersion were calculated for the quantitative variables. The normality of the knowledge level variable was tested with the Kolmogorov-Smirnov test. A *p* < 0.05 value was obtained for both the total knowledge level and for the three dimensions, so the null hypothesis of equality with respect to the normal distribution was rejected. Accordingly, non-parametric tests—Mann–Whitney’s U test and the Kruskal–Wallis test—were used to assess the statistical significance of group differences.

Additionally, ordinal logistic regression was used to estimate the effects of the explanatory variables pertaining to the level of knowledge, establishing a high level of knowledge (“High Knowledge”) as the reference category. Maximum theoretical models were generated that included all of the independent variables. The final models were calculated from these, and all of the significant variables were included. Subsequently, from the estimates of each variable included in the final model, the probability of occurrence of each category of the level of knowledge was calculated from the combinations of the categories of the predictor variables in the model. In order to determine the effect of each category of the predictor variables, a probability ratio (odds ratio) associated with each category was calculated for the three categories of the dependent variable.

The level of statistical significance was set at *p* < 0.05. The SPSS statistical package for Windows v.27 was used.

### 2.5. Ethical Considerations

The research ethics committee of the University of Jaen (Spain) approved the study protocol (MAR. 16/2). The participants voluntarily signed an informed consent form and received written information on the study. Furthermore, the researchers ensured the confidentiality of the data collected. 

## 3. Results

A hundred and thirty students participated in the study (104 were women). The mean age of the sample was 20.36 years (standard deviation [SD] = 1.535). Ninety percent of women and 77% of men had had sex at the time of the study, with no statistically significant difference (χ^2^ = 3.260; *p* = 0.071). The mean age of onset of sexual intercourse was 16.81 years (SD = 1.279). Table 1 shows the main characteristics of the sample.

The figures are expressed as percentages for the qualitative variables and as mean and standard deviation for the quantitative variables.

The mean for the overall knowledge level was 10.38 points (SD: 2.51); for knowledge regarding heterosexuality, 5.28 points (SD: 1.194); for knowledge regarding male condoms, 4.54 points (SD: 0.916); and for knowledge regarding hormonal contraceptives, 1.27 points (SD: 1.346).

At the bivariate level, no statistically significant differences were found for the dimension of knowledge about heterosexuality in any of the variables analyzed. For the dimension of knowledge about male condoms, those participants who wanted to use male condoms in the future had a higher knowledge level than those who planned to use the contraceptive pill or vaginal ring, but this difference was not statistically significant. The dimension of knowledge of the male condom was the only one where men had higher knowledge scores than women (4.62 vs. 4.52; *p* = 0.720). The main knowledge gap we observed was in the dimension of knowledge about hormonal contraceptives. Women who attended public schools were more knowledgeable than those who studied in other schools. Finally, at the global level, those who say they would use contraceptive pills in future sexual relations had significantly higher scores than the vaginal ring participants (Z = −2.536; *p* = 0.011) and the male condom participants, although this difference was not significant (Z = −1.654; *p* = 0.098). Table 2 shows the results of all the hypothesis contrasts at the bivariate level.

A re-coding of the dependent variable was performed to test the model using the ordinal logistic regression technique. Table 3 shows a high level of global knowledge about heterosexuality and contraception in 121 participants, while nine exhibited a level of knowledge coded as low. When the dimensions of heterosexuality and male condom knowledge were assessed, a high level of knowledge was observed for these dimensions. In contrast, 61 respondents exhibited low knowledge of hormonal contraceptive methods, 38 participants scored zero in this knowledge category, and only 31 showed high knowledge.

For the dimensions of knowledge about heterosexuality, knowledge about male condoms, and global knowledge, a binomial logistic regression model was used, as it has two categories of knowledge levels. In contrast, for the dimension of knowledge about hormonal contraception, an ordinal logistic regression model was used, establishing as reference category “High Knowledge” for all the models. The result of the analysis was that, for global knowledge, knowledge about heterosexuality, and knowledge about male condoms, no model was obtained that improved the proper adjustment of the constant, so it was concluded that it is not possible to carry out analyses of these dimensions beyond the dichotomous analysis. For the dimension of knowledge of hormonal contraception, we found that the null hypothesis of adaptation of the model only with the constant could be rejected (χ^2^ = 13.613; *p* = 0.018). Concerning the goodness-of-the-fit of the model, it obtained a value of *p* higher than a significance level of 0.05, so the null hypothesis cannot be rejected, which suggests that the data adapt to the model. Next, we observe that the PseudoR-Square value of the variability explained by the model is 12.9%. The variables proposed by the model were sex, school, use of contraception during the first sexual experience, and the contraceptive method of choice for future sexual relations. Finally, the parallel lines test was performed. The result was *p* = 0.083, so the null hypothesis cannot be rejected, indicating that the ordinal procedure is feasible for the data. From the parameter estimates, the odds ratios associated with each of the combinations of predictor variables were calculated (Table 4 and Table 5).

## 4. Discussion

For the development of a safe experience with heterosexuality during youth, it is necessary to have sufficient knowledge to be able to respond in the best way to possible doubts related to the use of the different contraceptive options and the correct use of each method according to the characteristics of each situation, thus preventing the occurrence of an unwanted pregnancy or sexually transmitted infection.

The results of this study show that the main source of information used to obtain information on sexuality and contraception is the Internet for both men and women. There is a change from previous studies with respect to the main source of information. The reason for this change, as suggested by Gómez et al. [22], may be a result of a sense of shame experienced by young people when talking to their parents and, in turn, experienced by parents in talking about sexuality-related issues. This sense of shame causes young people to turn to the Internet as an easy and quickly accessible tool for obtaining information. On the other hand, the loss of importance of the media as a source of information about sexuality may be caused by the evolution of technology, where the Internet has acquired great importance and is accessible from mobile phones. The studies selected are from 2000 [23] and 2005 [18,24], years in which Internet access and use were not as widespread as today. Our results coincide with subsequent research such as that of Rahman et al. [19], where the Internet is emerging as the main source of information for young university students and is likewise maintained as the main source in the survey conducted by the Spanish Society of Contraception [4], and in recent research on undergraduate nursing students [11,19].

With respect to the age of first sexual experience, our results coincide with previous research [11,12,24] that place it at around 16 years of age, being somewhat lower in males than in females, unlike in the present study (average age in males is 16.90 years of age = 1.373 and in females, it is 16.80 years of age = 1.265), but this difference is not significant (*p* = 0.742). A decrease in the age of sexual debut is found in the adolescent age group, highlighting the importance of implementing training in schools since 28% of adolescents and young people reported that they had not received information on sexuality and contraception at school. In Spain, education on sexuality and contraceptive methods is the responsibility of the health system and the autonomous community, with the contents and teaching methodologies varying from one educational center to another [12]. The training received generally focuses on the prevention of unwanted pregnancy from a heterosexual sexuality perspective. The training has a central focus on the male condom, with a gap in knowledge about hormonal contraceptive methods, as we can observe in our research. In order to achieve the objective proposed by the WHO [17] of reaching a sufficient body of knowledge, an early approach to training is necessary before adolescents begin sexual relations, which allows the development of attitudes and practices based on the best scientific knowledge available.

The male condom is the method of choice during the first sexual experience. These results coincide with other research [11,12,25,26] on university populations. Regarding the contraceptive method for future relations, the male condom remains the contraceptive method of choice for our respondents, followed by the contraceptive pill and the vaginal ring. 

In the present investigation, 93.1% of the participants presented a level of global knowledge, encompassing heterosexuality and contraception, coded as high, while 6.9% of the participants presented a low level of knowledge. Hernández-Martínez et al. [27] found that the majority of participants had a low level of knowledge. This variability of results may be due to variations in the standards selected to measure the level of knowledge. On the other hand, the participants in the present research were students enrolled in the second year of a Degree in Nursing, so as they all are pursuing a degree in health sciences, they may have a concern about sexuality and contraceptive methods that lead them to inform on their own, thus raising the level of knowledge.

With respect to the main knowledge gap identified in our research on hormonal contraceptives, this is consistent with the studies of Najure et al. [28], Maafo Darteh et al. [29], and y Aziken et al. [30], who also found a low level of knowledge on how to administer the pill, both the birth control pill and the emergency contraceptive pill.

Through our evaluation of the variables that influence the level of knowledge about hormonal contraceptive methods, four predictor variables were obtained. The calculated odds ratios indicated that the probability of having high knowledge is higher in women, in participants who studied in a public school, in those who did not use any contraceptive method during their first sexual experience, and among those who wish to use the contraceptive pills as a contraceptive method in future sexual intercourse. These results are consistent with research by Maafo Darteh et al. [29], in which women are 2.8 times more likely to know about emergency contraception. The differences in results according to the type of school can be explained by the fact that the private institutions were religious, and the students may not have had access to sexuality and contraceptive training. With respect to differences in contraceptive use at first sexual intercourse, the reason why participants did not use any contraceptive method should be explored, as this motivation may include differences in the level of knowledge about hormonal contraception. The contraceptive method used during first sexual experiences was the male condom for all participants who used contraception, and the differences in the level of knowledge found in the present research may be found in the mistrust of this contraceptive method and the motivation to seek alternative contraceptive methods such as hormonal methods for future sexual relations. Finally, we observed that when the category of wanting to use the contraceptive pill in future sexual relations is presented, the probability of having high knowledge is higher than that of having low or null knowledge (those with high knowledge were more likely to want to use contraceptive pills in the future than those with low or no knowledge). This may be due to participants who want to use this contraceptive method for future sexual relations being motivated to learn more about hormonal contraceptive methods.

The study has some strengths and weaknesses that need to be addressed when interpreting our findings. For the selection of independent variables, a systematic review was carried out to determine the main variables influencing the level of knowledge of young people. To analyze the effect of the variables, all of the variables were introduced into the regression model to analyze the effect of all the variables simultaneously and to select those that have a significant effect on the dependent variable analyzed.

Noteworthy limitations of this research are the difference in size between the groups (men and women) and the differences between those who are sexually active and those who are not, depending on the contraceptive method they wish to use in future relationships, so we should be cautious when making comparisons between the groups.

One aspect to highlight in future research is the assessment of the rate of contraceptive use in all sexual relations and whether the desire to use a particular contraceptive method is related to its use in future sexual relations. More research is needed to assess young people’s level of knowledge about sexuality and contraception.

The created model can explain 12.9% of the variability in hormonal contraception knowledge from the variables included in the model. For future research, we recommend looking for other variables that can improve the explanatory capacity of the model.

## 5. Conclusions

Nursing students have a high level of general knowledge about heterosexuality and contraception; however, there is still a gap in knowledge about hormonal contraceptives. Regarding the level of knowledge about hormonal contraceptive methods, differences were observed according to the sex of the participants, with higher scores for females, along with the participants who would prefer to use the contraceptive pill in their future sexual relationships. These results highlight the need to reinforce training in early adolescence. Therefore, the creation of specific training programs on sexuality and contraceptive methods in schools is required.

## Figures and Tables

**Table 1 healthcare-10-01695-t001:** Characteristics of the participants.

Variable	Categories	Man (n = 26)	Woman (n = 104)	Total (n = 130)
Age	Median (SD)	20.65 (1.548)	20.29 (1.543)	20.36 (1.535)
School	Public	88.5	87.5	87.5
	Private	11.5	12.5	12.3
Source of information used	Internet	73.1	59.6	62.3
	Friends	15.4	23.1	21.5
	Talks on sexual and reproductive health	7.7	9.6	9.2
	TV	3.8	7.7	6.9
Source of information demanded	Internet	34.6	31.7	32.3
	Educational brochures	38.5	39.4	39.2
	health professionals	15.4	22.1	20.8
	TV	11.5	6.7	7.7
Self-perception of knowledge	average	11.5	10.6	10.8
	Good	88.5	89.4	89.2
Sexual experience	Yes	89.4	76.9	86.9
	No	10.6	23.1	13.1
Age at first sexual experience	Median (SD)	16.90 (1.37)	16.80 (1.27)	16.81 (1.279)
Reason to have sex	Pleasure	36	23.7	25.7
	Love	60	73.1	70.8
	To feel accepted	5	3.2	3.5
Use of contraception during first sexual experience	Yes	95	92.5	92.9
	No	5	7.5	7.1
Contraceptive used during first sexual experience	Male condom	100	100	100
Contraceptive method of choice for future sexual relations	Male condom	80	80.6	80.5
	Contraceptive pills	15	12.9	13.3
	Vaginal ring	5	6.5	6.2

**Table 2 healthcare-10-01695-t002:** Bivariate contrast of the knowledge scale.

Knowledge about Heterosexuality			
Variable	Categories	Mean (SD)	Contrast
Sex	Male	5 (1.52)	Z = −0.871; *p* = 0.384
	Female	5.35 (1.09)	
School	Public	5.28 (1.20)	Z = −0.110; *p* = 0.912
	Private	5.25 (1.24)	
Age			Rho = −0.011; *p* = 0.901
Source of information used	Internet	5.16 (1.26)	χ^2^ = 7.70; *p* = 0.053
	Friends	5.50 (1.17)	
	Talks on sexual and reproductive health	5.33 (0.99)	
	TV	5.56 (0.88)	
Self-perception of knowledge	Good	5.26 (1.22)	Z = −0.272; *p* = 0.786
	Regular	5.43 (0.94)	
Sexual experience	Yes	5.33 (1.18)	Z = −1.515; *p* = 0.130
	No	4.94 (1.25)	
Age of first sexual experience			Rho = 0.041; *p* = 0.663
Use of contraception during first sexual experience	Yes	5.30 (1.21)	Z = −1.019; *p* = 0.308
	No	5.75 (0.71)	
Contraceptive method of choice for future sexual relations	Male condom	5.27 (1.21)	χ^2^ = 1.476; *p* = 0.478
	Hormonal pill	5.60 (1.12)	
	Vaginal ring	5.43 (0.98)	
**Knowledge about male condom**			
Variable	Categories	Mean (SD)	Contrast
Sex	Male	4.62 (0.80)	Z = −0.359; *p* = 0.720
	Female	4.52 (0.95)	
School	Public	4.54 (0.88)	Z = −0.179; *p* = 0.858
	Private	4.50 (1.15)	
Age			Rho = −0.021; *p* = 0.814
Source of information used	Internet	4.46 (1.01)	χ^2^ = 2.562; *p* = 0.464
	Friends	4.79 (0.63)	
	Talks on sexual and reproductive health	4.50 (0.91)	
	TV	4.56 (0.88)	
Self-perception of knowledge	Good	4.52 (0.94)	Z = −0.716; *p* = 0.474
	Regular	4.71 (0.73)	
Sexual experience	Yes	4.54 (0.89)	Z = −0.310; *p* = 0.757
	No	4.53 (1.13)	
Age at first sexual experience			Rho= −0.080; *p* = 0.400
Use of contraception during first sexual experience	Yes	4.50 (0.91)	Z = −1.555; *p* = 0.120
	No	5 (0.0)	
Contraceptive method of choice for future sexual relations	Male condom	4.60 (0.81)	χ^2^ = 2.712; *p* = 0.258
	Hormonal pill	4.47 (0.92)	
	Vaginal ring	3.86 (1.57)	
**Knowledge about hormonal methods**			
Variable	Categories	Mean (SD)	Contrast
Sex	Male	1.08 (1.20)	Z = −0.696; *p* = 0.486
	Female	1.33 (1.38)	
School	Public	1.39 (1.39)	Z = −2.566; *p* = 0.010 *
	Private	0.50 (0.52)	
Age			Rho= −0.041; *p* = 0.643
Source of information used	Internet	1.37 (1.355)	χ^2^ = 2.950; *p* = 0.399
	Friends	0.93 (0.89)	
	Talks on sexual and reproductive health	1.50 (1.83)	
	TV	1.22 (1.72)	
Self-perception of knowledge	Good	1.25 (1.33)	Z = −0.737; *p* = 0.461
	Regular	1.50 (1.45)	
Sexual experience	Yes	1.29 (1.37)	Z = −0.089; *p* = 0.929
	No	1.18 (1.13)	
Age at first sexual experience			Rho = 0.073; *p* = 0.442
Use of contraception during first sexual experience	Yes	2 (1.85)	Z = −1.532; *p* = 0.125
	No	1.24 (1.33)	
Contraceptive method of choice for future sexual relations	Male condom	1.26 (1.20)	χ^2^ = 6.921; *p* = 0.031 *
	Hormonal pill	1.87 (1.38)	
	Vaginal ring	0.43 (0.54)	
**Global knowledge**			
Variable	Categories	Mean (SD)	Contrast
Sex	Male	10.63 (1.59)	Z = −0.972; *p* = 0.331
	Female	11.92 (2.09)	
School	Public	11.21 (1.97)	Z = −1.334; *p* = 0.182
	Private	10.25 (2.11)	
Age			Rho= −0.064; *p* = 0.471
Source of information used	Internet	10.99 (2.15)	χ^2^ = 0.441; *p* = 0.932
	Friends	11.21 (1.25)	
	Talks on sexual and reproductive health	11.33 (2.46)	
	TV	11.33 (2.12)	
Self-perception of knowledge	Good	11.03 (2.03)	Z = −0.787; *p* = 0.431
	Regular	11.64 (1.69)	
Sexual experience	Yes	11.16 (2.01)	Z = −0.745; *p* = 0.456
	No	10.54 (20.3)	
Age at first sexual experience			Rho= 0.073; *p* = 0.442
Use of contraception during first sexual experience	Yes	11.04 (1.95)	Z = −2.209; *p* = 0.027
	No	12.75 (2.12)	
Contraceptive method of choice for future sexual relations	Male condom	11.43 (2.01)	χ^2^ = 6.131; *p* = 0.047 *
	Hormonal pill	11.93 (1.53)	
	Vaginal ring	9.71 (2.29)	

* *p* < 0.05.

**Table 3 healthcare-10-01695-t003:** Level of knowledge about heterosexuality and contraception.

Categories	Knowledge about Heterosexuality	Knowledge about Male Condom	Knowledge on Hormonal Contraception	Global Knowledge
High	93.8	98.5	23.8	93.1
Low	6.2	1.5	46.9	6.9
Null	0	0	29.2	0

Data expressed in percentages.

**Table 4 healthcare-10-01695-t004:** Estimated odds ratio for the sex and college variables.

Variables	Sex OR (High/Low) ^a^	Sex OR (High/Null) ^a^	School OR (High/Low) ^b^	School OR (High/Null) ^b^
Male/Public/No contraceptive use/male condom	1.82	2.86	2.75	9.09 **
Male/Private/No contraceptive use/male condom	1.47	2.78	-	-
Male/Public/Yes contraceptive use/Male condom	1.62	2.90	1.85	7.52 **
Male/Private/Yes contraceptive use/Male condom	1.25	2.48	-	-
Male/Public/No contraceptive use/Contraceptive pills	1.98	2.63	3.33	9.36 **
Male/Private/No contraceptive use/Contraceptive pills	1.69	2.93	-	-
Male/Public/Yes contraceptive use/Contraceptive pills	1.81	2.89	2.54	9.04 **
Male/Private/Yes contraceptive use/Contraceptive pills	1.46	2.76	-	-
Male/Public/No contraceptive use/Vaginal ring	1.68	2.93	2.03	8 **
Male/Private/No contraceptive use/Vaginal ring	1.32	2.56	-	-
Male/Public/Yes contraceptive use/Vaginal Ring	1.45	2.74	1.50	6.40 *
Male/Private/Yes contraceptive use/Vaginal ring.	1.15	2.31	-	-
Female/Public/No contraceptive use/Male condom	-	-	3.20	9.45 **
Female/Private/No contraceptive use/Male condom	-	-	-	-
Female/Public/Yes contraceptive use/Male condom	-	-	2.40	8.80 **
Female/Private/Yes contraceptive use/Male condom	-	-	-	-
Female/Public/No contraceptive use/Contraceptive pills	-	-	3.82 *	8.40 **
Female/Private/No contraceptive use/Contraceptive pills	-	-	-	-
Female/Public/Yes contraceptive use/Contraceptive pills	-	-	3.17	9.46 **
Female/Private/Yes contraceptive use/Contraceptive pills	-	-	-	-
Female/Public/No contraceptive use/Vaginal ring	-	-	2.63	9.15 **
Female/Private/No contraceptive use/Vaginal ring	-	-	-	-
Female/Public/Yes contraceptive use/Vaginal Ring	-	-	1.89	7.62 **
Female/Private/Yes contraceptive use/Vaginal ring.	-	-	-	-

^a^ Exposure variable Female/male; ^b^ Exposure variable Public/private school; *: *p* < 0.05; **: *p* < 0.01.

**Table 5 healthcare-10-01695-t005:** Estimated odds ratio for the variables contraceptive use during first sexual experience and contraceptive method of future choice.

Variables	Contraceptive Use OR (High/Low) ^a^	Contraceptive Use OR (High/Null) ^a^	Future Contraceptive Future OR(High/Low)	Future Contraceptive Used OR (High/Null)
Male/Public/No contraceptive use/male condom	1.90	3.95 *	1.61 ^b^	2.71 ^b^
Male/Private/No contraceptive use/male condom	1.37	3.27	1.28 ^b^	2.38 ^b^
Male/Public/Yes contraceptive use/Male condom	-	-	1.40 ^b^	2.55 ^b^
Male/Private/Yes contraceptive use/Male condom	-	-	1.13 ^b^	2.16 ^b^
Male/Public/No contraceptive use/Contraceptive pills	2.18	3.87 *	2.11 ^c^	3.65 ^c^
Male/Private/No contraceptive use/Contraceptive pills	1.66	3.74	1.63 ^c^	3.54 ^c^
Male/Public/Yes contraceptive use/Contraceptive pills	-	-	1.84 ^c^	3.73 ^c^
Male/Private/Yes contraceptive use/Contraceptive pills	-	-	1.34 ^c^	3.10 ^c^
Male/Public/No contraceptive use/Vaginal ring	1.65	3.72	3.41 ^d^	9.87 ^d^**
Male/Private/No contraceptive use/Vaginal ring	1.22	2.97	2.09 ^d^	8.44 ^d^**
Male/Public/Yes contraceptive use/Vaginal Ring	-	-	2.58 ^d^	9.50 ^d^**
Male/Private/Yes contraceptive use/Vaginal ring.	-	-	1.52 ^d^	6.71 ^d^**
Female/Public/No contraceptive use/Male condom	2.13	3.92 *	1.75 ^b^	2.66 ^b^
Female/Private/No contraceptive use/Male condom	1.60	3.66	1.43 ^b^	2.58 ^b^
Female/Public/Yes contraceptive use/Male condom	-	-	1.57 ^b^	2.69 ^b^
Female/Private/Yes contraceptive use/Male condom	-	-	1.24 ^b^	2.33 ^b^
Female/Public/No contraceptive use/Contraceptive pills	2.33	3.53	2.25 ^c^	3.33 ^c^
Female/Private/No contraceptive use/Contraceptive pills	1.93	3.97 *	1.88 ^c^	3.75 ^c^
Female/Public/Yes contraceptive use/Contraceptive pills	-	-	2.06 ^c^	3.71 ^c^
Female/Private/Yes contraceptive use/Contraceptive pills	-	-	1.56 ^c^	3.45 ^c^
Female/Public/No contraceptive use/Vaginal ring	1.92	3.96 *	3.93 ^d*^	8.86 ^d^**
Female/Private/No contraceptive use/Vaginal ring	1.38	3.30	2.70 ^d^	9.66 ^d^**
Female/Public/Yes contraceptive use/Vaginal Ring	-	-	3.23 ^d^	9.97 ^d^**
Female/Private/Yes contraceptive use/Vaginal ring.	-	-	1.93 ^d^	8.03 ^d^**

^a^ Exposure variable for contraceptive use at first sexual intercourse Not used/used; ^b^ Variable contraceptive exposure they wish to use in future sexual relations CC pill/Male condom; ^c^ Variable contraceptive exposure they wish to use in future sexual relations CC pill/Vaginal ring; ^d^ Variable contraceptive exposure they wish to use in future sexual relations Male condom/Vaginal ring; *: *p* < 0.05; **: *p* < 0.01.

## Data Availability

All data generated or analyzed during this study are included in this published article [Appendix A].

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
