# Peer review of "Knowledge of Sexuality and Contraception in Students at a Spanish University: A Descriptive Study"

_healthcare, 2022, doi:10.3390/healthcare10091695_

Round 1
Reviewer 1 Report
I am always keen to support the publication of good quality research originating from outside the dominant western, English-speaking world (i.e., UK, North America, Australasia) as research from other countries can potentially offer useful insights to the existing research corpus. While this paper does not offer anything highly original to the field it provides a valuable insight from a Spanish perspective, and (potentially) some important points of difference from similar research undertaken in other countries. However, the authors have not made the most of this. The research itself appears to be sound, and with some changes the paper would be publishable.
Specific points that need addressing are as follows:
1. As this paper is primarily about contraception, some clarity is needed around the term 'sexuality'. Given that the term sexuality is generally understood to encompass a much wider remit than is covered in this paper, I would recommend that the authors use '(hetero-)sexuality' in all instances.
2. The statement at lines 33-34 is problematic in that it implies that experimentation is primarily about sexual risktaking. While this is usually considered a form os experimentation, the literature on adolescent development discusses a range of forms of experimentation common at this point of the lifespan. This point therefore needs revising to indicate that this is just one form of experimentation.
3. The sentence beginning "In recent years..." (lines 35-37) presents a claim that appears to be specific to the Spanish context, but I am not sure is substantiated in the wider literature. Immediately prior to this, I would recommend beginning a new paragraph and giving an overview of what the wider literature says about age of initiation. My understanding is that in most contexts it has remained stable, so this needs to be articulated and accompanied by relevant citations. It would then be fine to present the claim at lines 35-37 but to contextualise it: e.g. "However, research undertaken in Spain found that...".
4. The word 'disinformation' is used at line 46. I am not sure this is the correct word. Check the difference between 'misinformation' and 'disinformation' and then amend if necessary.
5. The opening sentence of the final paragraph in the introduction (lines 53-55) needs contextualising. Add the words "in Spain" to the end of this sentence.
6. Lines 64 and 67 the phrase "2nd course" - should this be "2nd year course"? Also, write 2nd out in full (i.e., second).
7. Line 66 it should be "convenience sample" (not "sampling").
8. Lines 103-108 something is amiss here. I am not sure if these should bullet pointed or descriptive prose. Please revise.
9. Line 156 it should be "wanted" (not "wants")
10. Line 157 it should be "planned" (not "plants"). Also, add the word 'the' before 'contraceptive' (e.g., "the contraceptive pill").
11. Line 211 this statement is unclear - please revise so that it is more specific.
12. The statement at lines 216-218 suggests that young people not wanting to talk with parents about sex and contraception is a new thing. Is this the case? Research from the English-speaking world suggests that young people have felt embarrassed about talking with parents about these things (actually parents feel embarrassed about discussing these things with their kids too!). It used to be that peers were the main source of information, but this has now been superceded by the internet. Revise this paragraph to ensure that the claims made are both accurate and contextualised to the Spanish situation.
Author Response
|
Comment |
Response |
Page/line number |
|
1. As this paper is primarily about contraception, some clarity is needed around the term 'sexuality'. Given that the term sexuality is generally understood to encompass a much wider remit than is covered in this paper, I would recommend that the authors use '(hetero-)sexuality' in all instances. |
Modified the term sexuality to hetero-sexuality. Information about the term sexuality is added at the introduction section |
Page 2, line 64-72 |
|
2. The statement at lines 33-34 is problematic in that it implies that experimentation is primarily about sexual risktaking. While this is usually considered a form os experimentation, the literature on adolescent development discusses a range of forms of experimentation common at this point of the lifespan. This point therefore needs revising to indicate that this is just one form of experimentation. |
Added at lines 43 and 44. |
|
|
3. The sentence beginning "In recent years..." (lines 35-37) presents a claim that appears to be specific to the Spanish context, but I am not sure is substantiated in the wider literature. Immediately prior to this, I would recommend beginning a new paragraph and giving an overview of what the wider literature says about age of initiation. My understanding is that in most contexts it has remained stable, so this needs to be articulated and accompanied by relevant citations. It would then be fine to present the claim at lines 35-37 but to contextualise it: e.g. "However, research undertaken in Spain found that...". |
Added at lines 46-50 |
|
|
4. The word 'disinformation' is used at line 46. I am not sure this is the correct word. Check the difference between 'misinformation' and 'disinformation' and then amend if necessary. |
Modified |
Page 2, line 61 |
|
5. The opening sentence of the final paragraph in the introduction (lines 53-55) needs contextualising. Add the words "in Spain" to the end of this sentence. |
Added the contextualization |
Page 2, line 78 |
|
6. Lines 64 and 67 the phrase "2nd course" - should this be "2nd year course"? Also, write 2nd out in full (i.e., second). |
Modified |
Page 2, line 88-91 |
|
7. Line 66 it should be "convenience sample" (not "sampling"). |
Modified |
Page 2, line 90 |
|
8. Lines 103-108 something is amiss here. I am not sure if these should bullet pointed or descriptive prose. Please revise. |
Modified at methology section |
Page 3, line 108-144 |
|
9. Line 156 it should be "wanted" (not "wants") |
Modified |
Page 6, line 226 |
|
10. Line 157 it should be "planned" (not "plants"). Also, add the word 'the' before 'contraceptive' (e.g., "the contraceptive pill"). |
Modified |
Page 6, line 228 |
|
11. Line 211 this statement is unclear - please revise so that it is more specific. |
Modified |
Page 11, line 284-287 |
|
12. The statement at lines 216-218 suggests that young people not wanting to talk with parents about sex and contraception is a new thing. Is this the case? Research from the English-speaking world suggests that young people have felt embarrassed about talking with parents about these things (actually parents feel embarrassed about discussing these things with their kids too!). It used to be that peers were the main source of information, but this has now been superceded by the internet. Revise this paragraph to ensure that the claims made are both accurate and contextualised to the Spanish situation. |
Modified |
Page 11, line 292-296 |

Reviewer 2 Report
Introduction needs usto date information and resarch references. Need to explain claeary why the simply link between sex and reproduction. Why youth is the time for impulsiveness and risk-taking. This is a quite consertative and represive concepction of sexualiñty aproval by World Health Organization and you need a deep axplanation.
Lenaguaje used is old-fashion, unusual in modern sexuality research as "intercouse".
Deep confusion and mistakes that need also an explanation between lines 252 and 254 about knowlegde
Conclusions need more information. Explain why there is a lack of sexuality education in the universities and secondary schools.
Author Response
|
Introduction needs usto date information and resarch references. Need to explain claeary why the simply link between sex and reproduction. Why youth is the time for impulsiveness and risk-taking. This is a quite consertative and represive concepction of sexualiñty aproval by World Health Organization and you need a deep axplanation. |
Introduction modified according to WHO guidelines. |
Page 1-2, line 34-42; line 64-72 |
|
Lenaguaje used is old-fashion, unusual in modern sexuality research as "intercouse". |
Language has been modified accordingly |
|
|
Deep confusion and mistakes that need also an explanation between lines 252 and 254 about knowledge |
Modified |
Page 12, line 342-345 |
|
Conclusions need more information. Explain why there is a lack of sexuality education in the universities and secondary schools. |
Modified Lack of sexuality education in the universities and secondary schools explained in Discussion section |
Page 12, line 313-322 Page 13, line 386-393 |

Round 2
Reviewer 2 Report
Fine now